# THE SAMPLING-GAUSSIAN FOR STEREO MATCHING

## ABSTRACT

The *soft-argmax* operation is widely adopted in neural network-based stereo matching methods to enable differentiable regression of disparity. However, networks trained with *soft-argmax* tend to predict multimodal probability distributions due to the absence of explicit constraints on the shape of the distribution. Previous methods leveraged Laplacian distributions and cross-entropy for training but failed to effectively improve accuracy and even increased the network's processing time. In this paper, we propose a novel method called *Sampling-Gaussian* as a substitute for *soft-argmax*. It improves accuracy without increasing inference time. We innovatively interpret the training process as minimizing the distance in vector space and propose a combined loss of L1 loss and cosine similarity loss. We leveraged the normalized discrete Gaussian distribution for supervision. Moreover, we identified two issues in previous methods and proposed extending the disparity range and employing bilinear interpolation as solutions. We have conducted comprehensive experiments to demonstrate the superior performance of our *Sampling-Gaussian* method. The experimental results prove that we have achieved better accuracy on five baseline methods across four datasets. Moreover, we have achieved significant improvements on small datasets and models with weaker generalization capabilities. Our method is easy to implement, and the code is available online.

## 1 INTRODUCTION

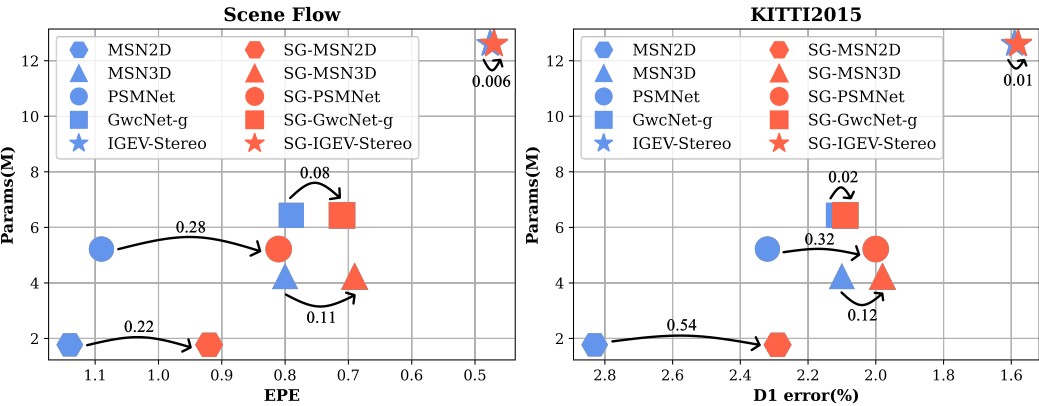

Figure 1: Quantitative comparisons on Sceneflow and Kitti. We implement our *Sampling-Gaussian* (SG) with five baseline methods for comparison. They are MSN2D and MSN3D (Shamsafar et al., 2021), PSMnet(Chang & Chen, 2018), GwcNet-g(Guo et al., 2019), IGEV-Stereo(Xu et al., 2023)

Stereo matching is a fundamental topic in computer vision that has been extensively researched for many years. Accurate stereo matching is essential for deriving scene depth, which is achieved by determining the displacement of corresponding points in binocular images. Stereo matching applications span a wide range of advanced technologies, including autonomous driving, robot navigation, and drone control.

The common baseline for end-to-end learning-based stereo matching, as described in (Mayer et al., 2016b), comprises three key modules: feature extraction, cost volume aggregation, and *soft-argmax*-

based disparity regression (Kendall et al., 2017a). Features are extracted from the input image pair via a siamese network architecture. Subsequently, a 5D cost volume $(B, C, D, H, W)$ is generated by concatenating features from the left and right images, with disparity as the additional dimension $D$. This cost volume then serves as input to a disparity regression module, which employs 3D convolutions to refine the output. Kendall et al. (2017a) was the first to leverage *soft-argmax* to achieve differentiable regression of disparity. Its efficiency and simplicity have made it a popular baseline for numerous subsequent studies (Chang & Chen, 2018; Pan et al., 2020; Wang et al., 2021; Xu et al., 2022; Shen et al., 2023). Various innovative modules have been proposed to improve stereo matching, such as feature fusion (Xu & Zhang, 2020; Guo et al., 2019), robust aggregation (Zhang et al., 2019a; Shamsafar et al., 2021), and iterative regression (Teed & Deng, 2021; Xu et al., 2023; 2024a). However, *soft-argmax* remains a key component of these methods.

As the cost volume passes through 3D CNNs, the number of channels is progressively reduced to 1. Subsequently, the *soft-argmax* module is applied to obtain the disparity map.

$$d = \sum_i i * softmax(z_i) = \sum_i i * \frac{e^{z_i}}{\sum e^{z_i}}. \tag{1}$$

$d$ denotes the predicted disparity. $i$ and $softmax(z_i)$ denotes the index of disparity and the probability of $i$.

$$smoothl1(d, \hat{d}) = \begin{cases} 0.5(d - \hat{d})^2, & if |d - \hat{d}| < 1 \\ |d - \hat{d}| - 0.5, & otherwise \end{cases}, \tag{2}$$

Then, a smooth L1 loss (Equation 2) is used to measure the distance between the predicted disparity $d$ and ground-truth $\hat{d}$. Since the *soft-argmax* function is widely adopted, researchers have also noticed its limitations. Kendall et al. (2017a) regarded the *soft-argmax* as a probability distribution of disparity and pointed out that it is prone to being influenced by multimodal distributions, as it estimates a weighted summation of all modes. Similarly, Chen et al. (2019) demonstrated that the predicted disparity of a multimodal distribution is deviated from the center of the dominating mode. They concluded that ambiguous matching is the cause of the multimodal problem. Researchers have proposed various methods aimed at solving this problem (Häger et al., 2021; Bangunharcana et al., 2021; Tulyakov et al., 2018; Xu et al., 2024b). These methods can be broadly summarized in two steps: constructing a direct supervision signal for the probability distributions to be predominantly unimodal, and limiting the disparity range of *soft-argmax* through post-processing.

It's challenging to reduce ambiguous matching relying solely on the network's regularization. Therefore, Tulyakov et al. (2018) constructed an explicit supervision signal based on a normalized discrete Laplacian distribution.

$$q(x) = \frac{1}{N} e^{\frac{-|x - \mu|}{2}}, \tag{3}$$

where $N = \sum_i e^{\frac{-|i - \mu|}{2}}$, $\mu$ is the ground-truth disparity and $q(x)$ is the probability of integer $x$. The learning process is supervised by a cross-entropy loss,

$$H(p, q) = \sum_{x \in [d_{min}, d_{max})} p(x) log(q(x)), \tag{4}$$

$p$ is the estimated probability. Inspired by their method, different distributions have been adopted, including Gaussian (Chen et al., 2019), Laplacian (Tulyakov et al., 2018; Xu et al., 2024b; Liu et al., 2021; Zhang et al., 2019b), and Dirac impulse Häger et al. (2021), etc. Distribution-based supervision effectively encourages the network to learn to estimate a distribution centered on the highest likelihood. However, post-processing, such as Top-k or equivalent processes, is still needed for multimodal distributions. Consequently, this results in an efficiency reduction because the operation is not parallelizable.

To address these issues, we propose a novel Gaussian distribution-based supervision method called *Sampling-Gaussian* as a substitute for *soft-argmax*. As shown in Figure 1, our method achieves significant improvements over the commonly used baselines listed. We provide a novel interpretation of disparity regression (Eq. 1) as a dot product between two vectors. Based on this interpretation, we leverage L1 loss and cosine similarity loss for optimization. Additionally, our method does not rely on any post-processing techniques. It can be directly applied to any *soft-argmax*-based stereo

matching algorithm without a decrease in efficiency. This paper is organized as follows: In Section 3, a theoretical analysis is provided to fundamentally explain the cause of the multimodal issues introduced by *soft-argmax* and why previous methods failed to achieve significant improvements. In section 4, we introduce the three main modules of *Sampling-Gaussian*, combination loss, extended disparity range, and bilinear interpolation. In the experimental section, we have implemented our method with five popular baselines(Chang & Chen, 2018; Shamsafar et al., 2021; Guo et al., 2019; Xu et al., 2023) to demonstrate that our method is easy to implement and universally applicable. At last, our method has also achieved state-of-the-arts results on Sceneflow(Mayer et al., 2016a) and Kitti2012, (Geiger et al., 2012), Kitti2015(Menze & Geiger, 2015), ETH3D(Schöps et al., 2017), and Middlebury(Scharstein et al., 2014).

In conclusion, our contributions has three folds:

- We propose *Sampling-Gaussian* as a substitute for *soft-argmax*. Our experiments demonstrate it's compatible with mainstream methods and requires minimal modifications to the original structures. Additionally, it improves accuracy without increasing processing time.

- We innovatively interpret *soft-argmax* (Eq. 1) from the perspective of vector space and propose a combination loss (Eq. 8) based on this interpretation. And disparity range extension and bilinear interpolation are proposed to address the unsolved issues of previous methods.

- We achieve significant improvements on small datasets and models with weaker generalization capabilities. Experiments on ETH3D, Middlebury, and MSN2D further validate our contributions.

## 2 RELATED WORKS

### 2.1 SOFT-ARGMAX-BASED METHODS

Based on the work of Kendall et al. (2017b), the subsequent improvement methods can be classified into several categories: feature level, module level, baseline level, and distribution level. Firstly, at the feature level, PSMnet(Chang & Chen, 2018) adopts a spatial feature pyramid(He et al., 2014) to fuse multi-resolution features, and stacked-hourglass module is adopted as regression module to improve the refinement. Guo et al. (2019) proposed a group-wise correlation network(GwcNet) for cost volume. Zhang et al. (2019a) proposed a guided-aggregation module to better refine the cost volume. At the baseline level, researchers proposed new baselines to improve the accuracy of the efficiency. Xu & Zhang (2020) and Pan et al. (2020) proposed to progressively aggregate the cost volume to the full size. Others proposed 2d convolution-based methods(Pan et al., 2024; Shamsafar et al., 2021) to reduce the high Flops. And Xu et al. (2023) proposed to iterative refine the disparity and significantly improve the accuracy but at the expense of speed.

### 2.2 DISTRIBUTION-BASED METHOD

The probabilities output by the softmax function can be interpreted as a probability distribution. Thus, the *soft-argmax* operation is equivalent to retrieving the mean of this probability distribution (Li et al., 2021). Consequently, networks trained with *soft-argmax* lack explicit supervision regarding the shape of the distribution, resulting in an unconstrained probability shape. Therefore, previous methods have not fully resolved the multimodal problem, prompting the development of various post-processing approaches to address this issue. PDS (Tulyakov et al., 2018) limit the range of the *soft-argmax* with Top-k during inference. Liu & Liu (2022) using learned weights to suppress unreliable disparity regions to increase the robustness. A similar idea was proposed in Häger et al. (2021), where they use a *Dirac impulse* to model the distributions.

## 3 EXPLORATIONS

In this section, we first analyze the biased gradient of *soft-argmax* to establish that distribution-based supervision is necessary for stereo matching. Then, we analyze the two basic settings that have caused previous distribution-based methods to their inferior improvements.

### 3.1 ANALYSIS OF BIASED GRADIENT

During the research, we observed that the input nodes, $e^{z_i}$, of the softmax function consistently receive biased gradients during backpropagation. Consequently, we conducted an analysis of this issue. The partial differential equation of *soft-argmax* (Eq.1) is,

$$
\begin{aligned}
\frac{\partial L}{\partial e^{z_i}} &= \frac{\partial L}{\partial d} \frac{\partial d}{\partial e^{z_i}} \\
&= \frac{\partial L}{\partial d} \left( i \frac{e^{z_i}}{\sum_* e^{z_*}} \left(1 - \frac{e^{z_i}}{\sum_* e^{z_*}}\right) + \sum_{j \neq i} j \left(-\frac{e^{z_j}}{\sum_* e^{z_*}} * \frac{e^{z_i}}{\sum_* e^{z_*}}\right) \right) \\
&= \frac{\partial L}{\partial d} \left( \frac{e^{z_i}}{\sum_* e^{z_*}} (i - d) \right).
\end{aligned}
\tag{5}
$$

The $e^{z_i} / \sum_* e^{z_*}$ denotes the normalized probability of the input node $e^{z_i}$, where $i$ represents the index of the nodes. Eq. 5 illustrates that the gradients received by $z_i$ during backpropagation are proportional to the distance $(i - d)$ between $i$ and $d$. As a result, the network receives biased gradients, preventing it from achieving optimal performance. We also believe this is the cause of the multimodal issue in *soft-argmax*.

### 3.2 ANALYSIS OF DISTRIBUTION-BASED METHOD

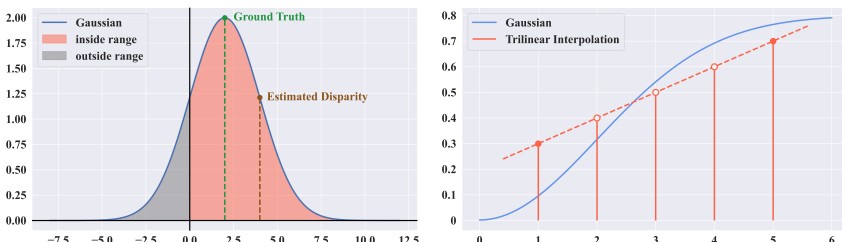

Figure 2: The *left* plot shows a truncated distribution near the endpoints, and its estimated disparity deviates from the ground truth. The *right* plot illustrates that the probabilities after trilinear interpolation are linearly distributed and cannot fit the Gaussian distribution well.

In the previous distribution-based, the *soft-argmax*(1) is interpreted as expectation of the network's predicted distribution. However, such methods fail to achieve good results for various reasons, and we believe there are two main reasons.

a) This disparity range is inherited from the *soft-argmax*-based method. As shown in left plot of Fig. 2, two issues arise with distribution-based methods. First, the generated distribution near the endpoints is truncated, causing the integration to be less than 1. Second, for models trained with such distributions, the expectation of their predicted distributions deviates from the ground truth. For instance, the distribution $q$ generated with ground truth near 0 as $\mu$, its expectation is larger than the full range one.

$$
\sum_{x=-\infty}^{\infty} x * q(x|\mu) < \sum_{x=0}^{\infty} x * q(x|\mu).
\tag{6}
$$

b) Trilinear interpolation is often used to upsample the feature map from $(D/4, H/4, W/4)$ to $(D, H, W)$. As shown in the right plot of Fig. 2, the upsampled probabilities on $D$-dimension are linearly distributed. However, the Gaussian distribution is not. Therefore, it's impossible for the network to learn the exact distribution. As a result, its expectation deviates from the ground truth.

## 4 THE PROPOSED *Sampling-Gaussian*

In this section, we present an innovative interpretation of the *soft-argmax* and disparity regression. Previous methods viewed the supervision process as minimizing the distance between two distributions, using L1 loss or cross-entropy loss for measurement. In our method, we view the the Eq. 1 as

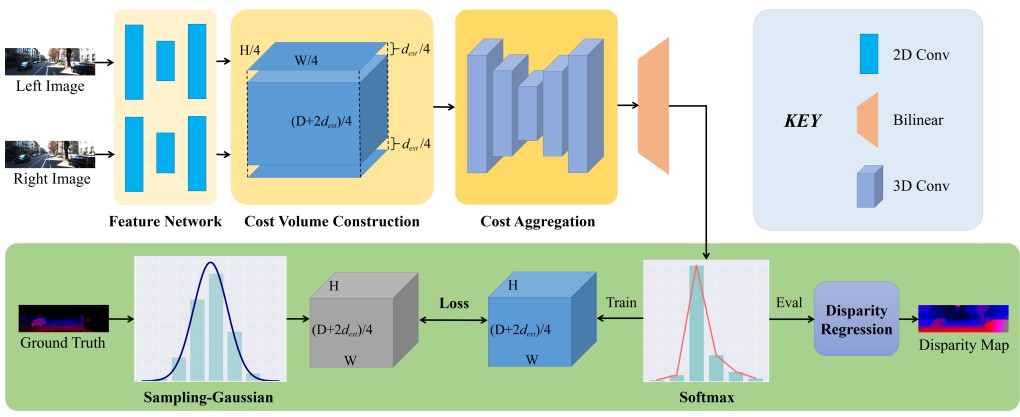

Figure 3: The workflow of our proposed *Sampling-Gaussian*

a dot product between two vectors, $i$ and $softmax(z_i)$. We construct a vector $q(i)$ such that $q(i) * i$ equals to ground truth. Since vector $i$ is always $[d_{min}, ..., d_{max}]^T$, minimizing the product between estimation and ground truth is equivalent to minimizing the distance between vectors $softmax(z_i)$ and $q(i)$. Based on this interpretation, we propose the *Sampling-Gaussian* method, which consists of three parts.

### 4.1 CONSTRUCT THE SUPERVISING SIGNAL

First, we extend the disparity range $D$ from $[0, d_{max})$ to $[-d_{ext}, d_{max} + d_{ext})$. Then we normalize the probability of the discrete Gaussian distribution within the extended range. The sampling function is defined as,

$$q(x) = \frac{e^{-\frac{(x-\mu)^2}{2\sigma^2}}}{\sum_x^{D/4} e^{-\frac{(x-\mu)^2}{2\sigma^2}}}. \tag{7}$$

The $\mu$ is the ground-truth disparity. $\sigma$ is used to control the shape, and $0.5$ achieves the best result.

### 4.2 COMBINATION LOSS

L1 loss is effective for measuring the distance between two vectors but lacks constraints on the angle between them. Two vectors with the same L1-norm can have very different dot products with $i$, as shown in Fig. 4. In response, we have proposed a combined loss of L1 and negative cosine similarity to measure both the L1-norm and the vectorial angle between vectors $p$ and $q$.

$$L(p, q) = \frac{1}{n} \sum_i^n |p(i) - q(i)| - \lambda * \frac{\sum_i^n p(i)q(i)}{\sqrt{\sum_i^n p(i)^2}\sqrt{\sum_i^n q(i)^2}}, \tag{8}$$

which the $\lambda = 0.5$ achieves the best performance based on our experiments.

### 4.3 BILINEAR INTERPOLATION

The cost volume constructed by fuse the features from left and right images. The construction of $C$ involves iteratively constructing the $C$ by shifting the feature map by $1$ pixel,

$$C(d, x, y) = g(f_l(x, y), f_y(x - d, y)). \tag{9}$$

The $f_l, f_r$ denotes the features of left and right image. And $g$ denotes a fusion method for features, usually is group-wise correlation(Guo et al., 2019) or concatenation(Chang & Chen, 2018). As shown in Fig.3, the size of $C$ is $[B, C, D/4, H/4, W/4]$. After the cost aggregation network, a *bilinear interpolation* is leveraged to upsample the cost volume after the regression modules,

$$\mathbf{C} = bilinear(C). \tag{10}$$

And size of $\mathbf{C}$ is $[B, C, D/4, H, W]$.

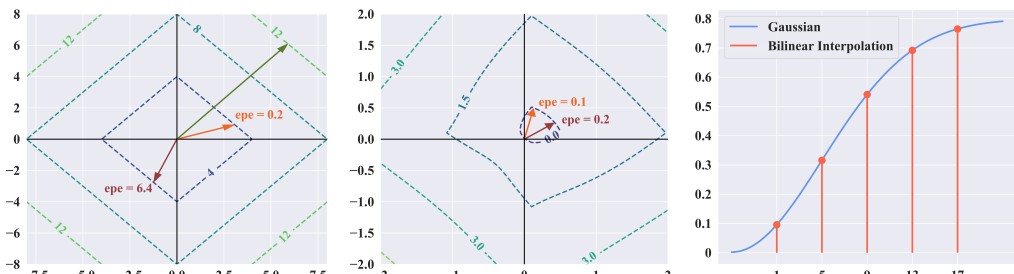

Figure 4: The *left* plot: The loss landscape of L1 loss, dashed lines are contour lines. Two vectors on the same contour line can have significant difference in endpoints error (epe) between their products and the ground truth. The *middle* plot: The loss landscape of the combined loss. Vectors on the same contour line have similar epes. The *right* plot: Since the predicted probabilities are not linearly related, they can fit into the Gaussian distribution.

### 4.4 INFERENCE

A key contribution of our method, is that we do not rely on post-processing operation for refinement. During the inference, we calculate the expectation of $p$ directly,

$$d = 4 * \sum_{i}^{D/4} i * p = 4 * \sum_{i}^{D/4} i * softmax(\mathbf{C}_i), \tag{11}$$

which has the same form of *soft-argmax*. Our method can be easily implemented with most of the *soft-argmax*-based method. The disparity range is $D/4$; consequently, the value $d$ after regression is also a quarter of its original value. Thus, the "$4*$" is used to recover $d$ to its full scale.

## 5 EXPERIMENTAL RESULTS

In this section, we report our implementation details and experimental results. We have implemented *Sampling-Gaussian* with 5 most representative methods for comparisons: 1. *PSMNet*(Chang & Chen, 2018). The "ResNet" of the stereo matching. Their method is open-source, easy to read and replicate. We use this method for a wider range of comparisons. 2. *GwcNet-g*(Guo et al., 2019). Their group-wise correlation module is also widely adopted, and their code is open-sourced. 3&4. *MSN3D* and *MSN2D* (Shamsafar et al., 2021): They have proposed lightweight networks by leveraging 2D convolutions to reduce computational expenses while maintaining accuracy. 5. *IGEV-Stereo*(Xu et al., 2023): A state-of-the-art (SOTA) method that adopts the iterative refinement module based on RAFT(Teed & Deng, 2021). We implement our method with IGEV-Stereo to demonstrate that our method is compatible with a variety of structures.

We conducted experiments on **four** datasets: **Sceneflow**(Mayer et al., 2016b) is a large scale of synthetic stereo dataset which contains more than 39k image pairs. **Kitti**(Geiger et al., 2012; Menze & Geiger, 2015), an open-road dataset contains 395 pairs for training and 395 pairs for testing. **ETH3D**(Schöps et al., 2017) is a gray-scale dataset with 27 training pairs and 20 testing pairs for a variety of scenes. **Middlebury**(Scharstein et al., 2014) is an indoor dataset, which provides 30 training pairs and testing pairs in three resolutions. We use the quarter-resolution for experiments.

### 5.1 IMPLEMENTATION DETAILS

For simplicity, we will refer to our *Sampling-Gaussian* as SG. Our implemented versions of method are denoted as SG-PSMNet or SG-MS2D. We conducted all the experiments on two A100 GPUs. We leverage AdamW(Loshchilov & Hutter, 2017) with $\beta_1 = 0.9, \beta_2 = 0.999$, weight decay$= 10^{-2}$, as optimizer. All the networks are trained with similar protocol: pretrain on Sceneflow for 20 epochs with lr$= 10^{-3}$. Then, finetuning on Kitti for 200 epochs with lr$= 10^{-3}$, then with lr$= 10^{-4}$ for another 300 epochs, and with lr$= 10^{-5}$ for the last 300 epochs. For IGEV-stereo and MSN2D, the parameters are slightly changed. Evaluation metrics(lower the better): *End-point error* (EPE)(Mayer

et al., 2016b), commonly used in optical flow. It calculates the l1 loss. *D1 error* (Menze & Geiger, 2015) calculates the percentage of error pixels. Pixels with EPE larger than 3 are considered as error.

## 5.2 ABLATION STUDIES

### 5.2.1 SIGMA $\sigma$ OF THE *Sampling-Gaussian*

Table 1: Quantitative comparisons on settings of $\sigma$

| $\sigma$ | 0.3 | 0.4 | **0.5** | 0.6 | 0.7 | 1.0 |
|---|---|---|---|---|---|---|
| PSMnet | 2.526 | 2.526 | **0.625** | 0.631 | 0.723 | 0.688 |

The $\sigma$ controls the shape of the distribution and directly affects the distribution pattern finally learned by the network. When $\sigma$ is set to 0.3 or 1, the shape of distribution is either too narrow or too wide. Either shape is hard for the network to learn which results in larger errors, as shown in table 1.

### 5.2.2 INTERPOLATION METHOD

Table 2: Quantitative comparisons on settings of $\sigma$.

| Base | Trilinear | Bilinear | Loss | $\lambda$ | EPE | D1 |
|---|---|---|---|---|---|---|
| MSN2D | ✓ | | L1 | / | 0.99 | 2.62 |
| | | ✓ | L1+Cos | 0.5 | **0.91** | **2.49** |
| PSMNet | ✓ | | CE | / | 0.94 | 2.34 |
| | ✓ | | L1 | / | 0.87 | 2.15 |
| | ✓ | | L1+Cos | 0.5 | 0.89 | 2.26 |
| | | ✓ | L1+Cos | 0.2 | 0.79 | 2.15 |
| | | ✓ | L1+Cos | 1.0 | 1.23 | 2.86 |
| | | ✓ | L1+Cos | 0.5 | **0.65** | **2.00** |

We have conducted experiments to compare bilinear interpolation with trilinear interpolation. As shown in table 2, bilinear interpolation has achieved better results with two methods, which aligns with our theory.

### 5.2.3 LOSSES AND LAMBDA $\lambda$

We have also conducted experiments to compare the performance of different combination of losses and weight $\lambda$. As shown in table 2, even though the cross-entropy(CE) loss has achieved only 0.94, the network converges faster than trained with L1 loss. Regarding the combination of L1 and Cosine similarity(Cos). Notably, if the $\lambda$ is set too large, the network would eventually collapse.

### 5.2.4 EXTENDED RANGE

Table 3: Ablation study on disparity range

| | Disparity Range | | | |
|---|---|---|---|---|
| | $(0, d_{max})$ | $(0, d_{max} + d_{ext})$ | $(-d_{ext}, d_{max})$ | $(-d_{ext}, d_{max} + d_{ext})$ |
| EPE | 0.425 | 0.415 | 0.396 | 0.389 |
| < 1 | 6.554 | 6.250 | 5.610 | 5.446 |
| < 3 | 0.787 | 0.785 | 0.741 | 0.676 |

In Sceneflow, points within the range of 0 to 16 accounts for 22.5% of the total, while the range of 176 to 192 accounts for 0.3%, resulting in a total of 22.8%. In KITTI, this range accounts for 16% of the total. Therefore, we conducted experiments on KITTI to evaluate the impact of extending the disparity range.

## 5.3 QUANTITATIVE COMPARISONS

Table 4: Quantitative comparison on Sceneflow

| Method | EPE | D1 | Params | Supervision | Loss | Top-k | Time(s) |
|---|---|---|---|---|---|---|---|
| PDS | 1.12 | 2.93 | 2.2 | Combined* | CE | Y | / |
| MSN2D | 1.14 | 2.83 | 2.23 | Soft-argmax | Smoothl1 | N | 0.10 |
| PSMNet | 1.09 | 2.32 | 5.22 | Soft-argmax | Smoothl1 | N | 0.41 |
| PSMNet+ | 1.02 | 3.12 | 2.32 | Laplacian | CE | Y | / |
| Acfnet | 0.87 | 4.31 | / | Combined* | CE+Focal | N | 0.48 |
| MSN3D | 0.80 | 2.10 | 1.77 | Soft-argmax | Smoothl1 | N | 0.53 |
| GwcNet-g | 0.79 | 2.11 | 6.43 | Soft-argmax | Smoothl1 | N | 0.32 |
| GANet+LaC | 0.72 | 6.52 | 9.43 | Combined* | L1+CE | Y | 1.72 |
| GANet+ADL | 0.50 | 1.81 | 9.43 | Laplacian | L1+CE | Y | 1.72 |
| IGEV-Stereo | 0.47 | 1.59 | 12.60 | Soft-argmax | L1 | N | 0.37 |
| SG-MSN2D | 0.91 | 2.49 | 2.23 | Gaussian | L1+Cos | N | 0.10 |
| SG-PSMNet | 0.65 | 2.00 | 5.22 | Gaussian | L1+Cos | N | 0.41 |
| SG-GwcNet-g | 0.71 | 2.09 | 6.43 | Gaussian | L1+Cos | N | 0.32 |
| SG-MSN3D | 0.69 | 1.98 | 1.77 | Gaussian | L1+Cos | N | 0.53 |
| SG-IGEV-Stereo | **0.47** | **1.58** | 12.60 | Gaussian | L1+Cos | N | 0.37 |

Combined*: combination of Soft-argmax and Laplacian

In this section, we compared with the SOTA methods and relative methods on Sceneflow, Kitti2012 and Kitti2015. In table 4, we compared with PDS(Tulyakov et al., 2018), Acfnet(Zhang et al., 2019b),PSMNet+(Chang & Chen, 2018), GANet+LaC(Liu et al., 2021), GANet+ADL(Xu et al., 2024b). Most distribution-based methods rely on post-processing modules for improvement, but this leads to an increase in latency. In contrast, our method effectively improves the accuracy of the baseline while keeping the architecture unchanged, thus ensuring consistent and efficient inference.

Table 5: The quantitative comparison on Kitti2012 and Kitti2015, the evaluation metrics are d1, $< 2$ and $< 3$ error rate(%). All are lower the better.

| Method | Kitti2015-All | | | Kitti2015-Noc | | | Kitti2012 | |
|---|---|---|---|---|---|---|---|---|
| | $d1_{bg}$ | $d1_{fg}$ | $d1_{all}$ | $d1_{bg}$ | $d1_{fg}$ | $d1_{all}$ | $< 2$ | $< 3$ |
| MSN2d(Shamsafar et al., 2021) | 2.49 | 4.53 | 2.83 | 2.29 | 3.81 | 2.54 | \ | \ |
| PDSNetTulyakov et al. (2018) | 2.29 | 4.05 | 2.58 | 2.09 | 3.68 | 2.36 | 4.65 | 2.53 |
| PSMnet(Chang & Chen, 2018) | 1.86 | 4.62 | 2.32 | 1.71 | 4.31 | 2.14 | 3.01 | 1.89 |
| PSMnet+CE(Chen et al., 2019) | 1.54 | 4.33 | 2.14 | 1.70 | 3.90 | 1.93 | 2.81 | 1.81 |
| GwcNet-g(Guo et al., 2019) | 1.74 | 3.93 | 2.11 | 1.61 | 3.49 | 1.92 | \ | \ |
| MSN3d(Shamsafar et al., 2021) | 1.75 | 3.87 | 2.10 | 1.61 | 3.50 | 1.92 | \ | \ |
| AAnet+(Xu & Zhang, 2020) | 1.65 | 3.96 | 2.03 | 1.49 | 3.66 | 1.85 | 2.96 | 2.04 |
| RAFT(Teed & Deng, 2021) | 1.48 | 3.46 | 1.81 | 1.34 | 3.11 | 1.63 | \ | \ |
| GANetZhang et al. (2019a) | 1.48 | 3.46 | 1.81 | 1.34 | 3.11 | 1.63 | 2.50 | 1.60 |
| ACVNet(Xu et al., 2022) | 1.37 | 3.07 | 1.65 | 1.26 | 2.84 | 1.52 | 2.34 | 1.47 |
| RT-IGEV++ (Xu et al., 2024a) | 1.48 | 3.37 | 1.79 | 1.34 | 3.17 | 1.64 | 2.51 | 1.68 |
| PSMNet+ADL(Xu et al., 2024b) | 1.44 | 3.25 | 1.74 | 1.30 | 3.04 | 1.59 | 2.17 | 1.42 |
| LEAstereoCheng et al. (2020) | 1.40 | 2.91 | 1.65 | 1.29 | 2.65 | 1.51 | 2.39 | 1.45 |
| IGEV-stereo(Xu et al., 2023) | 1.38 | 2.67 | 1.59 | 1.27 | 2.62 | 1.49 | 2.17 | 1.44 |
| SG-MSN2d | 1.94 | 4.07 | 2.29 | 1.78 | 3.63 | 2.08 | 3.15 | 2.09 |
| SG-GwcNet-g | 1.73 | 3.88 | 2.09 | 1.59 | 3.55 | 1.92 | 2.89 | 1.95 |
| SG-PSMnet | 1.77 | 3.13 | 2.00 | 1.65 | 2.97 | 1.87 | 2.69 | 1.80 |
| SG-MSN3d | 1.61 | 3.81 | 1.98 | 1.48 | 3.55 | 1.82 | 2.62 | 1.74 |
| SG-IGEV-stereo | 1.40 | **2.50** | **1.58** | 1.30 | **2.48** | 1.50 | **2.12** | **1.39** |

The comparisons on KITTI are listed in Table 5. Our method effectively improves the results of all baselines. Moreover, these results prove that our distribution model shows greater improvement for

those with weaker generalization abilities. Additionally, we achieved SOTA results with SG-IGEV-Stereo. In conclusion, *Sampling-Gaussian* effectively improves the generalization ability across a variety of model structures.

## 5.4 QUALITATIVE COMPARISONS

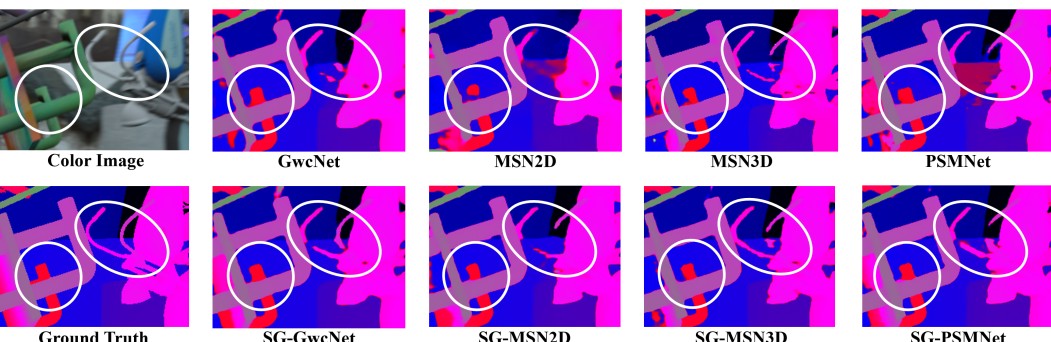

Figure 5: Qualitative comparisons on Sceneflow

Through experiments, we found that our *Sampling-Gaussian* effectively improves the accuracy of the model to predicts small objects and contours, as depicted in Fig. 5. The reason is that models trained with *Soft-argmax* are prone to converge to the majority of the disparity, while details are relatively in the minority. On the other hand, our SG provides explicit supervision for all objects. Therefore, the model gains the ability to capture details.

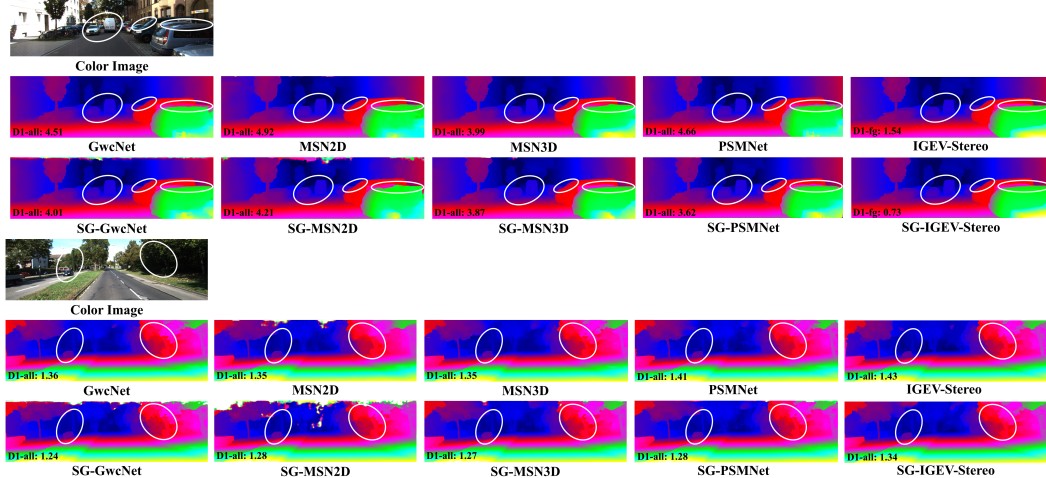

Figure 6: Qualitative comparisons on Kitti2015

In the first example in Fig. 6, it is evident that all baselines trained with SG have gained the ability to predict accurate contours of objects. For instance, in the disparity of the right side van and the shape of the trees in the background. More of our results are available on the Kitti2012 and Kitti2015 leaderboard.

## 5.5 EXPERIMENTS ON ETH3D AND MIDDLEBURY

ETH3D and Middlebury are both small datasets, each containing less than 30 samples. For a fair comparison, we divided the data with ground truth into training and validation sets. The results demonstrated that our method achieved significant improvements across nearly all approaches, highlighting its effectiveness, especially for small datasets.

Table 6: Quantitative comparisons on ETH3D and Middlebury

|  |  | MSN2D | | MSN3D | | PSMnet | | Gwc-g | |
|---|---|---|---|---|---|---|---|---|---|
|  |  | Base | SG* | Base | SG | Base | SG | Base | SG |
| ETH3D | EPE | 0.86 | **0.63** | 0.33 | **0.21** | 0.37 | **0.22** | 0.29 | **0.25** |
|  | D1 | 3.19 | **2.06** | 0.54 | **0.22** | 0.42 | **0.33** | 0.35 | **0.29** |
| Middlebury | EPE | 1.67 | **0.94** | 0.92 | **0.55** | 0.73 | **0.51** | 0.68 | **0.67** |
|  | D1 | 8.93 | **5.87** | 7.09 | **2.71** | 5.21 | **2.17** | **3.18** | 3.47 |

SG* : *Sampling-Gaussian*

## 5.6 CROSS-DOMAIN GENERALIZATION

Finally, we conducted experiments to evaluate the cross-domain generalization ability of our methods. We trained the baselines on Sceneflow and directly evaluated them on KITTI2015, ETH3D, and Middlebury. Our method demonstrated improved generalization performance across all three baselines. Qualitative results are available in appendix.

Table 7: Cross-domain generalization evaluation on Kitti2015, ETH3D and Middlebury

|  |  | Kitti2015 | | | ETH3D | | | Middlebury | | |
|---|---|---|---|---|---|---|---|---|---|---|
|  |  | EPE | $>1$ | $>3$ | EPE | $>1$ | $>3$ | EPE | $>1$ | $>3$ |
| MSN2D | Base | 5.03 | 56.1 | 24.4 | 7.24 | **18.46** | 9.38 | 5.95 | 41.0 | 18.1 |
|  | SG* | **1.53** | **48.2** | **12.5** | **3.71** | 18.82 | **6.17** | **1.67** | **31.3** | **15.7** |
| MSN3D | Base | 29.4 | 72.2 | 50.0 | 1.79 | 17.78 | 5.33 | 3.13 | 31.3 | 13.1 |
|  | SG | **22.5** | **53.7** | **17.3** | **1.66** | **8.03** | **4.32** | **2.60** | **26.5** | **11.4** |
| PSMnet | Base | **21.1** | 88.6 | **48.8** | 42.1 | 42.5 | 31.5 | 6.77 | 37.6 | 18.6 |
|  | SG | 24.6 | **78.0** | 57.2 | **5.40** | **14.1** | **5.40** | **6.07** | **29.3** | **15.1** |

SG* : *Sampling-Gaussian*

## 6 CONCLUSIONS

In this paper, we introduce a novel yet simple substitute for *soft-argmax*. Through comprehensive comparisons with five baseline methods, we demonstrate that our *Sampling-Gaussian* achieves improvements across a variety of model structures and datasets. Moreover, we propose a novel interpretation for distribution-based methods and introduce a combined loss function that achieves significant improvements. Additionally, we address the fundamental problems of previous distribution-based methods by extending the disparity range and employing bilinear interpolation. Lastly, our method proves effective for small datasets and models with weaker generalization abilities. In the future, we aim to study the generalization ability of stereo matching networks to enhance their applicability in real-life scenarios.

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

## A APPENDIX

### A.1 FULL EQUATION OF EQ. 5

The first part is the full equation of Eq. 5.

$$
\begin{aligned}
\frac{\partial L}{\partial e^{z_i}} &= \frac{\partial L}{\partial d}\frac{\partial d}{\partial e^{z_i}} \\
&= \frac{\partial L}{\partial d}(i\frac{e^{z_i}}{\sum_* e^{z_*}}(1 - \frac{e^{z_i}}{\sum_* e^{z_*}}) + \sum_{j \neq i} j(-\frac{e^{z_j}}{\sum_* e^{z_*}} * \frac{e^{z_i}}{\sum_* e^{z_*}})) \\
&= \frac{\partial L}{\partial d}(i\frac{e^{z_i}}{\sum_* e^{z_*}} + i(-\frac{e^{z_i}}{\sum_* e^{z_*}} * \frac{e^{z_i}}{\sum_* e^{z_*}}) + \sum_{j \neq i} j(-\frac{e^{z_j}}{\sum_* e^{z_*}} * \frac{e^{z_i}}{\sum_* e^{z_*}})) \\
&= \frac{\partial L}{\partial d}(i\frac{e^{z_i}}{\sum_* e^{z_*}} + \sum_j(-\frac{e^{z_j}}{\sum_* e^{z_*}} * \frac{e^{z_i}}{\sum_* e^{z_*}})) \\
&= \frac{\partial L}{\partial d}(\frac{e^{z_i}}{\sum_* e^{z_*}}(i - \underline{\sum_j j * \frac{e^{z_j}}{\sum_* e^{z_*}}})) \\
&= \frac{\partial L}{\partial d}(\frac{e^{z_i}}{\sum_* e^{z_*}}(i - d))
\end{aligned}
\tag{12}
$$

the part with underline is the equation of soft-argmax Eq. 1,

### A.2 PYTHON IMPLEMENTATION

This is the python implementation of *Sampling-Gaussian*.

```python
def groudtruth_to_gaussion(self, mean, sigma=0.5):
    gau_x = torch.Tensor(np.arange(-self.extra//4, (192+self.extra)//4)).unsqueeze(1).cuda()
    mean /= 4
    l = mean.shape[0]
    x = gau_x.repeat(1, l)
    ans = torch.exp(-1*((x-mean)**2)/(2*(sigma**2)))/(math.sqrt(2*np.pi)* sigma)
    ans /= torch.sum(ans,dim=0)
    return ans
```

### A.3 PROBABILITIES OF SAMPLING-GAUSSIAN

Table 8: The accuracy of the Sampling-Gaussian's cumulative possibility and expectation.

| $\mu$ | $1 - \sum_x p$ | $\mu - \sum_x d * p$ |
|---|---|---|
| 4 | 0.005296 | $-0.37134$ |
| 5 | 0.004317 | $-0.02964$ |
| 6 | $3.14e-05$ | $-0.00178$ |
| 7 | $1.10e-06$ | $-6.2e-05$ |
| 8 | $2.37e-08$ | $-1.3e-06$ |
| 7 | $1.10e-06$ | $-6.2e-05$ |
| 8 | $2.37e-08$ | $-1.3e-06$ |
| 9 | $3.07e-10$ | $-1.8e-08$ |
| 10 | $2.39e-12$ | $-1.4e-10$ |
| 11 | $1.09e-14$ | $-6.8e-13$ |
| 12 | 0.00 | $7.10e-15$ |
| 15 | 0.00 | 0.00.0 |
| 20 | 0.00 | $7.10e-15$ |

Let's review the equation 7. First, the probability density function of the discretized Gaussian distribution is defined as

$$q(x) = \frac{1}{\sigma * \sqrt{2\pi}} e^{-\frac{(x-\mu)^2}{2\sigma^2}} \tag{13}$$

The Riemann sum of the equation 13 is

$$\int_a^b e^{-\frac{(x-\mu)^2}{2\sigma^2}} dx \approx \frac{1}{2}(f(x_0) + 2f(x_1) \cdots + 2f(x_{N-1}) + f(x_N)) \tag{14}$$

We further evaluate the summation of probability of Eq. 14. Thus, we need to evaluate the *Sampling-Gaussian*'s cumulative possibility. As shown in Table 8. The table shows, that the cumulative possibility is not strictly equals to 1. However, the probabilities predicted by the network is strictly equals to 1 due to the softmax operation. Therefore, in Eq. 7, the probabilities is divided by the summation of the probabilities. Thus, the summation is strictly equals to 1.

The table 8 shown the range inside the $[0, d_{max})$. Which illustrate the reason of why $d_{ext}$ is needed. Moreover, as depicted in table 8. The cumulative possibility is not always equals to 1. Therefore, the division by the summation of the probabilities is an effective to strictly restrict the probability equals to 1.

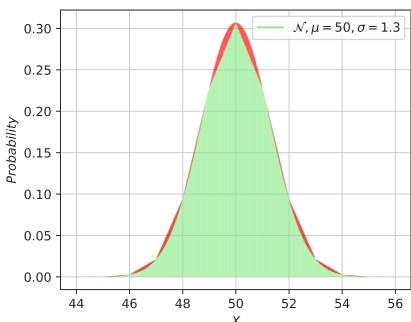

Figure 7: The green region represents the integral of Eq. 13, while the red area denotes the difference between the integrals and cumulative probability of $SG$.

### A.4 MORE ANALYSIS AND PROPERTIES

During the research, we have discovered that our *Sampling-Gaussian* possesses two interesting properties: Firstly, within a certain range of $\sigma \in [0.9, 1.7]$, its sum approximates to 1. Secondly, its expectation is equal to $\mu$.

The first property: that a finite integration of Gaussian distribution is defined by $\int_a^{a+1} e^{-\frac{(x-\mu)^2}{2\sigma^2}} dx$. The numerical integration is

$$\int_a^{a+1} e^{-\frac{(x-\mu)^2}{2\sigma^2}} dx \approx \frac{1}{2}(e^{-\frac{(a-\mu)^2}{2\sigma^2}} + e^{-\frac{(a+1-\mu)^2}{2\sigma^2}}). \tag{15}$$

Let $\{x_k\}$ be a partition of $[a, b]$, $a = x_0 < x_1 \cdots < x_{N-1} < x_N = b$, and the partition has a regular spacing $x_k - x_{k-1} = 1$. The approximation formula can be simplified as $\int_a^b e^{-\frac{(x-\mu)^2}{2\sigma^2}} dx \approx \frac{1}{2}(f(x_0) + 2f(x_1) \cdots + 2f(x_{n-1}) + f(x_n))$. Let $a = -\infty$, $b = \infty$, then we have

$$\frac{1}{\sigma\sqrt{2\pi}} \int_{-\infty}^{\infty} e^{-\frac{(x-\mu)^2}{2\sigma^2}} dx \approx \frac{1}{\sigma\sqrt{2\pi}} \sum_{x\in\mathbb{Z}} e^{-\frac{(x-\mu)^2}{2\sigma^2}}. \tag{16}$$

Second property: For simplicity, let $f(x) = e^{-\frac{(x-\mu)^2}{2\sigma^2}}$. $\forall x > \mu, \partial f/\partial x < 0$. Let $0 \leqslant t \leqslant 1, i < j$, $\forall x \in \{x_i | x \geqslant b, x_i \in \mathbb{Z}\}$, $f(x)$ satisfies $f(x_i + t*(x_j - x_i)) \leqslant f(x_i) + t[f(x_j) - f(x_i)]$. Therefore,

the numerical integration $\frac{1}{2}(x_n - x_1) \cdot (f(x_i) + f(x_n)) = \epsilon$ satisfies $\epsilon > \sum_{x=b}^{\infty} f(x) > 0$. Based on our numerical analysis, when $\delta = 5$, $\epsilon < 10^{-5}$, the

$$\frac{1}{\sigma\sqrt{2\pi}} \sum_{x \in \mathbb{Z}} f(x) - 2\epsilon = \frac{1}{\sigma\sqrt{2\pi}} \sum_{x=\mu-b}^{\mu+b} f(x) \approx 1. \tag{17}$$

Let $\mu \in (0, d_{max})$, $\sigma \in [0.5, 1.0]$, the expectation

$$E(x|\mu) = \sum_{x=0}^{d_{max}} \frac{1}{\sigma\sqrt{2\pi}} e^{-\frac{(x-\mu)^2}{2\sigma^2}} \approx \mu. \tag{18}$$

let $\mu \in (5, d_{max} - 5)$, $x^* \in \{x^* < 0 \cup x^* \geqslant d_{max}\}$. Then $E(x^*|\mu) \approx 0$. Given the finite range of disparity $[0, d_{max})$, by subtracting the $E(x^*|\mu)$ from the $E(x)$. We have also conducted experiments to quantize the error of the expectations and the error ranges from $10^{-5}$ to $10^{-12}$.

### A.5 TRAINING AND INFERENCE

The training and inference process is illustrated as:

---
**Algorithm 1** Training with **sampling-Gaussian**

---
**Input:** left, right image $I_l, I_r$, ground truth $\hat{d}$, sampling-Gaussian $f$, threshold $T$, set $S_x$.
**Output:** Network $N$.
  1: **while** $loss > T$ **do**
  2:     $y \leftarrow N(I_l, I_r)$
  3:     $d \leftarrow Softmax(y)$
  4:     $\hat{d} \leftarrow f(x = S_x | \mu = \hat{d})$
  5:     $loss \leftarrow L1(d, \hat{d}) - 0.5 * cos(d, \hat{d})$
  6:     update network by backpropagation
  7: **end while**

---

### A.6 THE RESULTS ON KITTI2012 AND KITTI2015

We provide the URL of our submitted results on Kitti leaderboard. SG-PSMNet on Kitti2015, SG-MSN2D on Kitti2015, SG-MSN3D on Kitti2015, SG-GwcNet-g on Kitti2015, SG-IGEV on Kitti2015. SG-PSMNet on Kitti2012, SG-MSN2D on Kitti2012, SG-MSN3D on Kitti2012, SG-IGEV on Kitti2012.

### A.7 THE CROSS-DOMAIN EXPERIMENTS ON ETH3D AND MIDDLEBURY

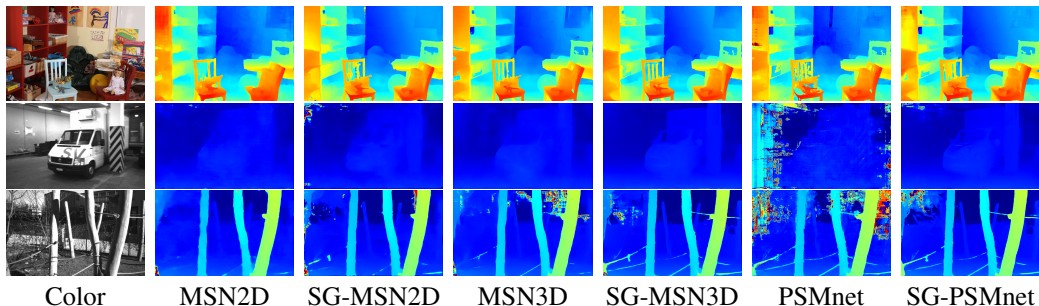

| Color | MSN2D | SG-MSN2D | MSN3D | SG-MSN3D | PSMnet | SG-PSMnet |

Figure 8: Quality comparisons on ETH3D and Middlebury of MSN2D, MSN3D, PSMnet and SG-MSN2D, SG-MSN3D, SG-PSMnet. The results demonstrate that our method exhibits better adaptability to different datasets in cross-domain experiments and ensures accurate estimation of object edges.

### A.8 MORE QUANTITATIVE COMPARISONS

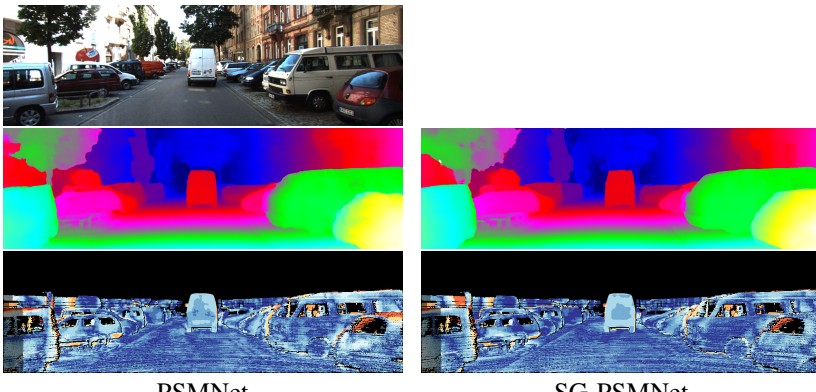

PSMNet                    SG-PSMNet

Figure 9: ALL-D1$_{bg}$, ALL-D1$_{fg}$, ALL-D1$_{all}$ are PSMNet: $(3.67, 1.16, 3.45)$, SG-PSMNet: $(3.24, 1.49, 3.08)$

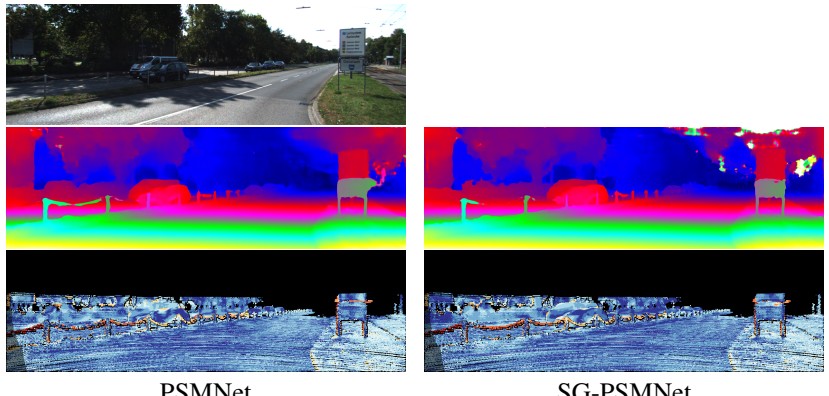

PSMNet                    SG-PSMNet

Figure 10: ALL-D1$_{bg}$, ALL-D1$_{fg}$, ALL-D1$_{all}$ are PSMNet: $(1.96, 2.22, 1.99)$, SG-PSMNet: $(1.66, 0.93, 1.58)$

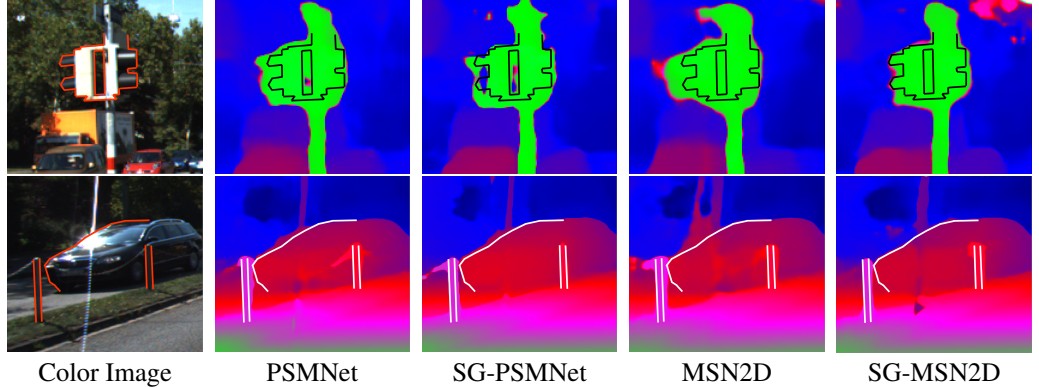

Color Image        PSMNet       SG-PSMNet       MSN2D       SG-MSN2D

Figure 11: Qualitative comparisons on Kitti2015. We manually marked the outline of the objects for better illustration.

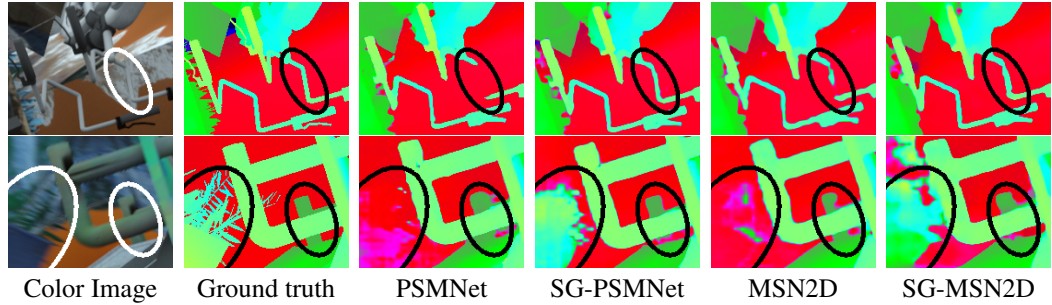

| Color Image | Ground truth | PSMNet | SG-PSMNet | MSN2D | SG-MSN2D |

Figure 12: Qualitative comparisons on Sceneflow.

