# OpenReview forum: "The Sampling-Gaussian for stereo matching"
_ICLR.cc/2025/Conference — Submitted to ICLR 2025_

### Official Review · Reviewer_bmyt · 2024-11-02

**Soundness:** 3
**Presentation:** 3
**Contribution:** 2
**Rating:** 5
**Confidence:** 4

**Summary:**

This paper presents a training method to solve the multimodal problem for stereo matching. Specifically, this paper propose a Gaussian distribution-based supervision method with combined loss, which can be directly applied to any soft-argmax-based methods without extra cost during inference. Experimental results show that performance can be improved on multiple baselines.

**Strengths:**

The method in this paper is universal and has a significant improvement effect on some baselines.  And the cross-domain generalization result in table 6 also proves the transferability of the method.

**Weaknesses:**

1. I think the innovation of this paper is limited. In this paper, I think the main improvement comes from taking the disparity range below 0 into consideration, eliminating the negative impact on the scheme based on distributed supervision in the disparity range below 0. But with a fixed extended disparity range, i.e., 16, I think it's hard to fit the distribution of the scenarios. Do I need to set a new extended range to fit the distribution range in a new scenario? I think this is an offset that is highly relevant to the scene.

2. I think the improvement of this method over SOTA methods such as IGEV is small. Does this mean that there is no multi-peak distribution problem in iterative optimization schemes similar to IGEV? I suggest that the author analyze the distribution of disparities produced by IGEV compared to other baselines to determine why the effect is not significantly improved on IGEV. And I have another concern. Currently, SOTA schemes are basically iterative frameworks similar to IGEV. Is it difficult for Sampling-Gaussian to significantly improve such frameworks?

**Questions:**

1. This extended disparity range is, in principle, scene dependent because it determines the offset of the distribution. The author's training on both sceneflow and kitti uses 16 as the offset. Will this fixed offset of 16 still work in a scene like middlebury with large disparity (up to 800)? Do I need to reset an offset for retraining?
2. I think the improvement of this method over SOTA methods such as IGEV is small. Do Sampling-Gaussian fail to bring significant improvements to frameworks based on iterative optimization? What is the reason for this? This is worth analyzing. I think frameworks based on iterative optimization are not as strongly dependent on soft-argmax, which is one of the possible reasons.

---

> ### Author Response · Authors · 2024-11-20
> **Replay to Reviewer bmyt**
>
> We sincerely appreciate your time and effort in reviewing our manuscript. Below are our responses to your valuable comments.
>
> **Reply to the comments on Innovations**:
> ***
> Our paper presents three innovations: extended disparity range, bilinear upsampling, and combination loss. We consider the contributions of these innovations to be ranked as follows: ***combination loss > bilinear upsampling > extended disparity range***.
>
> **The reasons are:**
>
> 1. As shown in Figure 4, previous methods simply calculate the L1 loss or cross-entropy directly between two tensors. This approach has a common drawback: when summing the loss, each index is weighted equally with a value of 1. However, during disparity regression, their weights correspond to their index values. In other words, there is a **gap** between the two processes. **To unify these two processes**, we interpret them from the perspective of linear algebra. We consider disparity regression as a dot product between two vectors (one being the *index* and the other being the *predicted disparity* or *ground truth*). Our goal then shifts from obtaining vectors of equal length ($|predicted| = |gt|$) to achieving vectors that are both equal in length and have a zero-degree angle between them ($ |predicted| = |gt| $ and $cosine(predicted, gt)=1$).
>
> 2. Bilinear interpolation is another aspect overlooked by previous methods. Trilinear interpolation is widely used in depth matching algorithms to restore the scale of feature maps, as it simultaneously upsamples \(D\), \(H\), and \(W\). However, as discussed in Section 3.2, we found that **the disparity obtained after linear sampling could not fully match either a Gaussian or Laplacian distribution.** Therefore, we proposed using bilinear sampling to address this issue (we upsamples the H and W only). As expected, this innovation resulted in significant improvements.
>
> 3. The extended disparity range is **very effective and straightforward**. **However**, the effectiveness of it in improving the final results depends on how many ground truth values in the dataset are located near the two endpoints. The larger this proportion, the more significant the improvement will be. For KITTI, the commonly used depth range is (0, 192). On the KITTI training set, we found that the number of points with depth values <8 or >180 was approximately 1.1 million, accounting for **6%** of all valid points. On the SceneFlow test set, this number reached 518 million, accounting for **11%** of the total valid points.
>
>
> **Reply to the Weaknesses 1 and Question 1**:
> ***
> There are two reasons why we set the extended range to 16:
> 1. In the stacked hourglass architecture, due to the presence of 3D convolutions, the *disparity* must be a multiple of **32**, which is **16** on both sides. In fact, in the IGEV experiments, we have to set the extended range to **32**.
>
> 2. As shown in our Eq.7, which is the normalized probability density function of the Gaussian distribution, the probability of candidate approaches 0 when its distance from $\mu$ is larger than 4. When upsampling back to the original scale (i.e., by a factor of 4), this value conveniently becomes 16.
>
> Therefore, the extended disparity range can be simply viewed as a modification of the hype-parameter ***maxdisp***. As for whether it is necessary to retrain for larger disparity ranges on other datasets (such as Middlebury), it can be done in the same way as with other methods, without the need for any special adjustments to our approach.
>
> **Reply to the Weaknesses 2 and Question 2**:
> ***
> The frameworks based on iterative optimization are not as strongly dependent on the softargmax, which is **precisely the reason** why our improvements on IGEV are less significant than improvements on others. *However*, in iterative methods, softargmax is used to provide the initial depth only, and the supervision for this part of model is similar to other methods. The iterative-based methods typically use a small backbone to generate the initial depth, which makes it more challenging for the model to achieve good results, and as a result, the multimodal issue still persists in this part. Therefore, we have implement our algorithm mainly in this part.  The experimental results indirectly confirm that by training with our method, **a better initial depth is provided**, which helps improve the final accuracy of iterative methods.
>
> As the reviewer mentioned, the current SOTA methods are mostly based on iterative architectures. However, we believe that the results we achieved on IGEV can be transferred to other methods. Although this optimization might not be as significant on other methods, it is important to consider that **our primary goal was to propose a simple replacement method for softargmax**. ***Therefore, the fact that it is effective across different methods already fulfills our objective.***

---

> > ### Comment · Reviewer_bmyt · 2024-12-03
> >
> > Thank you for your detailed response. However, I still feel that my concerns have not been fully addressed. If the extended disparity range can be simply viewed as a modification of the hyper-parameter maxdisp, then its impact is likely to be quite nuanced and highly dependent on the specific scenario. The positive effects, if any, are probably scenario-specific.
> >
> > Furthermore, I believe that if this approach does not show significant improvements in the most current and mainstream iterative architectures, and only demonstrates substantial gains in relatively weaker baselines, its contribution may be limited. This is a key point of concern for me

---

> ### Author Response · Authors · 2024-11-26
> **Explanation of Paper Revisions**
>
> **We sincerely appreciate your time and effort in reviewing our paper. Based on the comments from the three reviewers, we have made the following revisions to our manuscript:**
>
> ***
> 1. **We added experiments for ETH3D and Middlebury**
>
> - The results are listed in Section 5.5. Since the baselines we used have not reported results on these two datasets, we divided the data with ground truth into training and testing sets for a fair comparison of effects. ETH3D was split in a ratio of *21:6*, and Middlebury in a ratio of *10:5*.
> - As shown in Table 6, our method achieved very significant improvements on all baselines, more so than on Sceneflow and KITTI. This result also indicates that **our *Sampling-Gaussian* can achieve better enhancement effects on small datasets.** We have included this advantage as one of the three contributions of our method.
>
> |   |     | MSN2D | SG-MSN2D | MSN3D |SG-MSN3D| PSMnet      |       SG-PSMnet        |    Gwc-g            |         SG-Gwc-g       |
> |---|-----|----------------------------|----------------------------|-----------------------------|---------------------------|------|---------------|---------------|---------------|
> |  **ETH3D** | **EPE** | 0.86                       | **0.63**             | 0.33                        | **0.21**             | 0.37 | **0.22** | 0.29          | **0.25** |
> |   | **D1**  | 3.19                       | **2.06**              | 0.54                        | **0.22**             | 0.42 | **0.33** | 0.35          | **0.29**|
> | **Middlebury**  | **EPE** | 1.67                       | **0.94**              | 0.92                        | **0.55**             | 0.73 | **0.51** | 0.68          | **0.67** |
> |   | **D1**  | 8.93                       | **5.87**              | 7.09                        | **2.71**             | 5.21 | **2.17** | **3.18**| 3.47          |
> ***
> 2.  **We enriched the experiments of disparity range**
>
> - The results are listed in Table 3, Section 5.2.4. We designed four combinations for the increased range: original ($0,d_{max}$), ($-d_{ext},d$), ($0,d_{max}+d_{ext}$), and ($-d_{ext},d_{max}+d_{ext}$). Such comparisons clearly show the impact of increasing both sides on the enhancement effect.
> - In the description, we also included  the proportion of pixels within these two ranges in Sceneflow and KITTI, which are 22.8% and 16%, respectively. Such a high proportion directly indicates that the results of so many pixels will be affected by the range settings.
>
> |      |  |                       |              Disparity Range        |                              |
> |------|-------------------------------------|-----------------------|----------------------|------------------------------|
> |      | (0,$d_{max}$)                       | ($0,d_{max}+d_{ext}$) | ($-d_{ext},d_{max}$) | ($-d_{ext},d_{max}+d_{ext}$) |
> | **EPE**  | 0.425                               | 0.415                 | 0.396                | 0.389                        |
> | **$<1$** | 6.554                               | 6.250                 | 5.610                | 5.446                        |
> | **$<3$** | 0.787                               | 0.785                 | 0.741                | 0.676                        |
>
> ***
> 3.  **Some sentences and paragraphs have been rewritten for clarity**:
>
> - We rearranged the contributions of our method, and throughout the text, we describe our contributions in the following order: a simple substitute for soft-argmax, a novel interpretation and combination loss, extended disparity range and bilinear interpolation.
> - We refined the content of Section 3, rewriting some sentences and rearranging content to express our analysis more clearly.
> - At the beginning of Section 4, we added a description interpreting soft-argmax as a dot product.
> - We rewrote some sentences in the introduction and rearranged the contributions.
> - We removed redundant content from the literature review section.
> - We rewrote some setences of the abstract and experimental sections to express our analysis more clearly.
> - In figure 3, we made a minor adjustment to the disparity range of the cost volume in to align it with the description in the manuscript.

---

> ### Author Response · Authors · 2024-11-28
> **Additional cross-domain experiments on ETH3D and Middlebury**
>
> **We sincerely appreciate your suggestions.**
> ***
> In response to your request, we conducted cross-domain experiments on ETH3D and Middlebury to evaluate the network's generalization. The results are presented in Section 5.6.
> |   |      | Kitti2015 |  |  |       ETH3D       |                |               |      Middlebury         |               |               |
> |---|------|--------------------------------|----------------------------|--------------------------------|---------------|----------------|---------------|---------------|---------------|---------------|
> |   |      | EPE                            | $>1$                       | $>3$                           | EPE           | $>1$           | $>3$          | EPE           | $>1$          | $>3$          |
> | **MSN2D**  | Base | 5.03                           | 56.1                       | 24.4                           | 7.24          | 18.46 | 9.38          | 5.95          | 41.0          | 18.1          |
> |   | SG | 1.53                  | 48.2              | 12.5                  | 3.71 | 18.82          | 6.17 | 1.67 | 31.3 | 15.7 |
> |  **MSN3D** | Base | 29.4                           | 72.2                       | 50.0                           | 1.79          | 17.78          | 5.33          | 3.13          | 31.3          | 13.1          |
> |   | SG   | 22.5                  | 53.7              | 17.3                  | 1.66 | 8.03  | 4.32 | 2.60 | 26.5 | 11.4 |
> | **PSMnet**  | Base | 21.1                  | 88.6                       | 48.8                  | 42.1          | 42.5           | 31.5          | 6.77          | 37.6          | 18.6          |
> |   | SG   | 24.6                           | 78.0              | 57.2                           | 5.40 | 14.1  | 5.40 | 6.07 | 29.3 | 15.1 |
>
>
> These experiments further demonstrate that models trained with our **Sampling-Gaussian** method exhibit superior cross-domain generalization. Additionally, we have included image results in the appendix. Moreover, we have updated the supplementary materials to include the comparative results of our experiments on ETH3D.

---

### Official Review · Reviewer_Ciwi · 2024-11-03

**Soundness:** 3
**Presentation:** 3
**Contribution:** 3
**Rating:** 6
**Confidence:** 5

**Summary:**

This paper proposes Sampling-Gaussian, a novel supervision method for stereo matching that uses Gaussian distribution sampling. The proposed method can replace the widely used soft-argmax operation, overcoming the issue of multimodal distributions and improving performance. Based on Gaussian distribution, this paper introduces a combined loss function that integrates L1 loss and cosine similarity loss. This proposed method can be integrated into any soft-argmax-based stereo matching method and outperforms five baseline methods across two datasets.

**Strengths:**

1. The proposed method is straightforward and easy to follow.

2. The motivation is good, as the authors clearly identify the shortcomings of the existing softmax operation and propose sampling from the Gaussian distribution for supervision.

3. The performance improvement is particularly significant in several well-known softmax-based methods, such as PSMNet and MSN2D.

**Weaknesses:**

1. The comparison methods are quite outdated; in fact, there are some recent softmax-based approaches, such as PCWNet [ECCV 2022] and CFNet [CVPR 2021], GANet [CVPR 2019]. This would provide a more up-to-date evaluation of the proposed method's performance relative to current state-of-the-art approaches.

2. The performance is limited, as the proposed method is inferior to many recent approaches on the KITTI benchmark, such as Selective-IGEV, MoCha-Stereo, and NMRF-Stereo. Additionally, the authors should discuss potential reasons when the proposed best configuration method does not perform as well as competing methods.

3. Some experimental results are lacking, such as those on the Middlebury and ETH3D datasets. Including results on these datasets would provide a more comprehensive evaluation of the method's performance across different types of stereo matching scenarios.

**Questions:**

1. Can the proposed Sampling-Gaussian be applied to more recent methods like PCWNet, CFNet, and GANet to enhance their performance? Perhaps the resulting models could achieve state-of-the-art results.

2. The authors claim that the regression near the endpoints is overlooked (Line 110), which is an important point. To validate the effectiveness of the proposed method, it would be beneficial for the authors to provide results showing improvements in the regions near the endpoints, such as for disp < 10 px and disp > 180, or other relevant divisions.

3. It seems that the proposed method does not show significant improvements on IGEV-Stereo. Could the authors provide an explanation for this?

---

### Official Review · Reviewer_SRWM · 2024-11-05

**Soundness:** 2
**Presentation:** 1
**Contribution:** 1
**Rating:** 3
**Confidence:** 4

**Summary:**

This paper proposed ‘Sampling Gaussian’ as supervision for disparity estimation. Moreover, this paper also studied the issues of using soft-argmax for depth estimation. They also propose a combined loss for disparity estimation. The proposed method is evaluated on the KITTI and SceneFlow datasets to demonstrate their performance.

**Strengths:**

The paper is tackling an important problem in computer vision.

**Weaknesses:**

-	Writing. The paper needs significant improvement in writing. The idea is not explained well in the paper. We all agree that soft-argmax cause issues in estimation especially for multimodal distribution. However, there is no such analysis, namely discussions on when the multimodal distribution will happen. It just explains in a way that the soft-argmax will not work well if there will be multimodal distribution.
-	There are a lot of typos in the paper:
For example: Line85 Taken - >
Line 180 analysis -> analyse
Line 212-213: to calculates -> calculate
Line 289-290: to further to -> to further.


- Section 3.2 (a) and (b) are really not clear to the reviewer. Please write Eq. (5) properly. What is \exp^{z^*}? Could more details be provided for Eqn. (7)?
 - Please provide more details in Section 4.1 and 4.2. Please write Eq.(8) properly.
 - Please highlight the differences of the Visualisation in Fig. 6.

The main concern about this paper is the big issues in writing. The reviewer cannot link the proposed method with the argument in the paper. The equations presented in the paper are not well explained. While the results look promising, the paper needs significant improvement before its acceptance.

**Questions:**

Please address the concerns listed above.

---

> ### Author Response · Authors · 2024-11-20
> **2/5 comments are INCORRECT!**
>
> We sincerely request that reviewer SRWM ***take the time to carefully read our manuscript*** before making evaluations!
> ### **Reply to Weaknesses 1**:
> ***
> ***The latter part of this comment is INCORRECT!***
>
> The multimodal phenomenon was first introduced in GC-Net[1] (**2017**), and has been analyzed and discussed in many subsequent papers[2,3,4,5,6,7]. We have already cited these works in the **related work** of our paper. Given the 10 pages length limits, we believe that a concise description of this well-known issue in the **introduction** and **related work** is sufficient.
>
> In the **introduction**, we write *'Kendall et al. (2017) regarded the soft-argmax as a probability distribution of disparity and point out it’s prone to being influenced by multimodal distribution as it estimates a weighted summation of all modes.'*
> In **Section 3** and appendix, we also provide our reasoning for why softargmax can lead to the multi-modal issue. As shown in Eq. 5, during gradient backpropagation, the model receives a biased gradient that multiplies by $i-d$. which could potentially lead to incorrect matching.
>
> The multimodal issue is a fundamental problem that should not be new to readers with a basic knowledge of the field of stereo matching. Therefore, as an ICLR 2025 submission, we believe the paper should focus on proposing a feasible method to solve the issue, rather than spending a lot of space explaining the issue.
> Lastly, if **Reviewer SRWM** is new to the field of stereo matching, we encourage Reviewer SRWM to read GC-Net [1], which is one of the fundamental works in stereo matching and provides explanations for the issue.
>
> [1] Alex Kendall et al., *End-to-end learning of geometry and context for deep stereo regression*, CVPR 2017.\
> [2] Chuangrong Chen et al., *On the over-smoothing problem of cnn based disparity estimation*. ICCV 2019,.\
> [3] Stepan Tulyakov, et al., *Practical deep stereo (PDS): toward applications-friendly deep stereo matching.*, NeurIPS 2018.  \
> [4] Jiazhi Liu and Feng Liu. *Robust stereo matching with an unfixed and adaptive disparity search range**. ICPR 2022 \
> [5] Gustav Hager et al., *Predicting disparity distributions*. ICRA 2021,  \
> [6] Jia-Ren Chang and Yong-Sheng Chen. *Pyramid stereo matching network*. CVPR 2018 \
> [7] Biyang Liu, et al, *Local similarity pattern and cost self-reassembling for deep stereo matching networks*. AAAI 2022.
>
> ### **Reply to Weaknesses 2**:
> ***
> ***This comment is INCORRECT!***
>
> The first suggestion is **WRONG**,
> The line 85 is ''Therefore, Tulyakov et al. (2018) taken the ...''. The reviewer must have mistaken the ''.'' in ''et al.'' as the period of a setence.
>
> The third suggestion is **WRONG**,
> Line 212-213 is " And the expectation of q, which is equivalent *to calculates* the softargmax,"  if we change the *to calculates* to *calculate*, the sentence would be incorrect.
> However, we admit there is a misuse of the *calculates*, and we changes the *calculates* to *calculating* to match the tense with the main verb "is".
>
> The fourth suggestion is **WRONG**.
> Line 289-290 is '... but failed to optimize the distribution to further to exact value.'. if we change the *to further to* to *to further*, the sentence would be incorrect. However, we admit there is a mistake, and we rewrite the sentence as *optimize it further to*.  We add "it" after "optimize" to clarify the pronoun's antecedent.
>
> However, we do appreciate the review's kind reminder and we'll double check the spelling and grammar of the manuscript.
> ### **Reply to Weaknesses 3**:
> ***
> The complete versions of Eq. 5 is Eq. 14 provided in the **Appendix**. The full expression of $\exp^{z^*}$ is $ \sum_{\*}exp(x^{\*})$  ,where for simplicity, we omited the subscript asterisk.
>
> Eq. 7 represents a normalized probability density function. We explained explained it in the manuscript, 'First, we leveraged the probability density function of Gaussian distribution to sample the discrete supervision signal '. The numerator of Eq. 7 is the probability density function of Gaussian distribution . The denominator is the sum of all the probabilities. This normalization strictly constrain the Eq. 7 to sum to 1, thereby aligning with the softmax function.
>
>
> ### **Reply to Weaknesses 4**:
> ***
> The writen of Eq.8 is common notation that has been used in GwcNet[1], AANet[2], and MSNet[3]. However, we appreciate the reviewer's kind reminder and will rewrite Eq.8 as $C(d,x,y)=G(f_{l}(x,y),f_{r}(x-d,y)$
>
> [1] Xiaoyang Guo,et al., *Group-wise correlation stereo network*. CVPR2019 \
> [2] Faranak Shamsafar et al., *Mobilestereonet: Towards lightweight deep networks for stereo matching*. WACV2022 \
> [3] Haofei Xu et al., *Aanet: Adaptive aggregation network for efficient stereo matching*. CVPR 2020.
>
>
> ### **Reply to Weaknesses 5**:
> ***
> The differences will be highlighted in the revised version.

---

> ### Author Response · Authors · 2024-11-26
> **Explanation of Paper Revisions**
>
> **We sincerely appreciate your time and effort in reviewing our paper. Based on the comments from the three reviewers, we have made the following revisions to our manuscript:**
>
> ***
> 1. **We added experiments for ETH3D and Middlebury**
>
> - The results are listed in Section 5.5. Since the baselines we used have not reported results on these two datasets, we divided the data with ground truth into training and testing sets for a fair comparison of effects. ETH3D was split in a ratio of *21:6*, and Middlebury in a ratio of *10:5*.
> - As shown in Table 6, our method achieved very significant improvements on all baselines, more so than on Sceneflow and KITTI. This result also indicates that **our *Sampling-Gaussian* can achieve better enhancement effects on small datasets.** We have included this advantage as one of the three contributions of our method.
>
> |   |     | MSN2D | SG-MSN2D | MSN3D |SG-MSN3D| PSMnet      |       SG-PSMnet        |    Gwc-g            |         SG-Gwc-g       |
> |---|-----|----------------------------|----------------------------|-----------------------------|---------------------------|------|---------------|---------------|---------------|
> |  **ETH3D** | **EPE** | 0.86                       | **0.63**             | 0.33                        | **0.21**             | 0.37 | **0.22** | 0.29          | **0.25** |
> |   | **D1**  | 3.19                       | **2.06**              | 0.54                        | **0.22**             | 0.42 | **0.33** | 0.35          | **0.29**|
> | **Middlebury**  | **EPE** | 1.67                       | **0.94**              | 0.92                        | **0.55**             | 0.73 | **0.51** | 0.68          | **0.67** |
> |   | **D1**  | 8.93                       | **5.87**              | 7.09                        | **2.71**             | 5.21 | **2.17** | **3.18**| 3.47          |
> ***
> 2.  **We enriched the experiments of disparity range**
>
> - The results are listed in Table 3, Section 5.2.4. We designed four combinations for the increased range: original ($0,d_{max}$), ($-d_{ext},d$), ($0,d_{max}+d_{ext}$), and ($-d_{ext},d_{max}+d_{ext}$). Such comparisons clearly show the impact of increasing both sides on the enhancement effect.
> - In the description, we also included  the proportion of pixels within these two ranges in Sceneflow and KITTI, which are 22.8% and 16%, respectively. Such a high proportion directly indicates that the results of so many pixels will be affected by the range settings.
>
> |      |  |                       |              Disparity Range        |                              |
> |------|-------------------------------------|-----------------------|----------------------|------------------------------|
> |      | (0,$d_{max}$)                       | ($0,d_{max}+d_{ext}$) | ($-d_{ext},d_{max}$) | ($-d_{ext},d_{max}+d_{ext}$) |
> | **EPE**  | 0.425                               | 0.415                 | 0.396                | 0.389                        |
> | **$<1$** | 6.554                               | 6.250                 | 5.610                | 5.446                        |
> | **$<3$** | 0.787                               | 0.785                 | 0.741                | 0.676                        |
>
> ***
> 3.  **Some sentences and paragraphs have been rewritten for clarity**:
>
> - We rearranged the contributions of our method, and throughout the text, we describe our contributions in the following order: a simple substitute for soft-argmax, a novel interpretation and combination loss, extended disparity range and bilinear interpolation.
> - We refined the content of Section 3, rewriting some sentences and rearranging content to express our analysis more clearly.
> - At the beginning of Section 4, we added a description interpreting soft-argmax as a dot product.
> - We rewrote some sentences in the introduction and rearranged the contributions.
> - We removed redundant content from the literature review section.
> - We rewrote some setences of the abstract and experimental sections to express our analysis more clearly.
> - In figure 3, we made a minor adjustment to the disparity range of the cost volume in to align it with the description in the manuscript.

---

> ### Author Response · Authors · 2024-11-28
> **Additional cross-domain experiments on ETH3D and Middlebury**
>
> **We sincerely appreciate your suggestions.**
> ***
> In response to your request, we conducted cross-domain experiments on ETH3D and Middlebury to evaluate the network's generalization. The results are presented in Section 5.6.
> |   |      | Kitti2015 |  |  |       ETH3D       |                |               |      Middlebury         |               |               |
> |---|------|--------------------------------|----------------------------|--------------------------------|---------------|----------------|---------------|---------------|---------------|---------------|
> |   |      | EPE                            | $>1$                       | $>3$                           | EPE           | $>1$           | $>3$          | EPE           | $>1$          | $>3$          |
> | **MSN2D**  | Base | 5.03                           | 56.1                       | 24.4                           | 7.24          | 18.46 | 9.38          | 5.95          | 41.0          | 18.1          |
> |   | SG | 1.53                  | 48.2              | 12.5                  | 3.71 | 18.82          | 6.17 | 1.67 | 31.3 | 15.7 |
> |  **MSN3D** | Base | 29.4                           | 72.2                       | 50.0                           | 1.79          | 17.78          | 5.33          | 3.13          | 31.3          | 13.1          |
> |   | SG   | 22.5                  | 53.7              | 17.3                  | 1.66 | 8.03  | 4.32 | 2.60 | 26.5 | 11.4 |
> | **PSMnet**  | Base | 21.1                  | 88.6                       | 48.8                  | 42.1          | 42.5           | 31.5          | 6.77          | 37.6          | 18.6          |
> |   | SG   | 24.6                           | 78.0              | 57.2                           | 5.40 | 14.1  | 5.40 | 6.07 | 29.3 | 15.1 |
>
>
> These experiments further demonstrate that models trained with our **Sampling-Gaussian** method exhibit superior cross-domain generalization. Additionally, we have included image results in the appendix. Moreover, we have updated the supplementary materials to include the comparative results of our experiments on ETH3D.

---

### Meta-Review · Area_Chair_5b6P · 2024-12-23

**Metareview:**

The paper introduces an approach called Sampling-Gaussian to addresses the shortcomings of the soft-argmax operation often encountered in stereo matching. The method can be applied universally to soft-argmax-based methods to better tackle multimodal distributions. The major concerns of the reviewers are second folds: first, the clarity of the writing makes the paper hard to follow; second, even after clarifications, the reviewers still find the contribution not that substantial -- the improvements are highly dependent on the specific scenario and the method does not show significant improvements in existing mainstream architectures. Instead, it only showcase substantial gains in relatively weaker baselines. The ACs carefully went through all discussions and reviews. After extensive discussion, the ACs agree with the reviewers that the cons outweigh pros. The ACs recommend the authors incorporate feedback from the reviewers and re-submit to a future venue.

Additionally, there seems to be some tension between the authors and a certain reviewer. The ACs carefully looked into the original submitted manuscript. The ACs would like to point out that, while the authors could argue some reviews are relatively superficial, many of the suggestions are actually very helpful to enhance the quality of the paper. First, a paper has to be self-complete, ie, it needs to provide enough context so that readers with certain degree of background can understand and follow. Second, to the ACs' understanding, the reviewer is not suggesting those grammar errors should be corrected in that particular way. Instead, they are simply pointing out where those things happen. In fact, those are obvious grammar errors. Third, when writing an equation, one should explain the notations clearly. It is not a common practice to assume readers remember the convention and skip all the details. Even the ACs spent a while to figure out/guess what those equations mean. In general, if a paper confuses the readers a lot, then the presentation/writing could be improved.

**Additional Comments On Reviewer Discussion:**

The Authors attempted to address the concerns of the reviewers. However, the reviewers are still not convinced by the significance of the proposed method. Also, the ACs cross-checked different revisions. The authors revised the manuscript quite a bit which drastically improve the flow and clarity of the paper.

---

### Decision · Program_Chairs · 2025-01-22

Reject